# Toward Multi-Database Query Reasoning for Text2Cypher

Makbule Gulcin Ozsoy[†]

[1]*Neo4j, London, UK*

## Abstract

Large language models have significantly improved natural language interfaces to databases by translating user questions into executable queries. In particular, Text2Cypher focuses on generating Cypher queries for graph databases, enabling users to access graph data without query language expertise. Most existing Text2Cypher systems assume a single preselected graph database, where queries are generated over a known schema. However, real-world systems are often distributed across multiple independent graph databases organized by domain or system boundaries, where relevant information may span multiple sources. To address this limitation, we propose a shift from single-database query generation to multi-database query reasoning. Instead of assuming a fixed execution context, the system must reason about (i) relevant databases, (ii) how to decompose a question across them, and (iii) how to integrate partial results. We formalize this setting through a three-phase roadmap: database routing, multi-database decomposition, and heterogeneous query reasoning across database types and query languages. This work provides a structured formulation of multi-database reasoning for Text2Cypher and identifies challenges in source selection, query decomposition, and result integration, aiming to support more realistic and scalable natural language interfaces to graph databases.

## Keywords

Text2Cypher, Multi-database query reasoning, Natural language interfaces to databases

## 1. Introduction

Large language models (LLMs) have enabled natural language interfaces to databases, where user questions are translated into formal query languages. This paradigm has been widely studied in Text2SQL, Text2SPARQL, Text2GQL, and Text2Cypher [1, 2, 3, 4]. In particular, Text2Cypher focuses on translating natural language into Cypher queries for graph databases, enabling users to access graph data without query language expertise.

Most existing Text2Cypher systems assume a single preselected graph database as the execution context and generate queries within it. However, real-world systems are often distributed across multiple independent databases organized by domain or system boundaries. In such environments, the information required to answer a single question may span multiple data sources, making the single-database assumption restrictive. This also introduces a source selection challenge, where the system must identify which database(s) are relevant from a set of candidates [5, 6]. Beyond query generation, the system must decide which sources are relevant, whether the question should be decomposed, how to decompose it when needed, and how to integrate partial results into a final answer. While prior work in relational settings has explored schema-level decomposition, schema-level retrieval and open-domain database retrieval [7, 5], they still assume a single target database or a unified schema. This assumption does not hold in multi-graph environments, where databases are independent and do not share a global structure. We illustrate this setting using a movie production company with separate HR, finance, and movies databases in Figure 1. The figure shows that each database captures a different aspect of the system, with no explicit links between them. To address these challenges, we propose a shift from single-database query generation to multi-database query reasoning. In this setting, the system must jointly perform source routing, query decomposition, and result integration in addition to query generation. We structure this problem through a three-phase roadmap:

*GenAIK-NORA: The Joint Workshop on Generative AI and Knowledge Graphs and KNOwledge GRaphs & Agentic Systems Interplay, 2026*

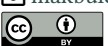 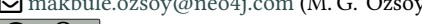 makbule.ozsoy@neo4j.com (M. G. Ozsoy)

- **Phase 1: Database Routing.** Given multiple graph databases and a natural language question, the system first selects the most relevant database and then generates a Cypher query over it.
- **Phase 2: Multi-Database Decomposition.** For questions spanning multiple databases, the system decomposes the question into sub-questions, assigns them to appropriate databases, generates per-database queries, and integrates the results.
- **Phase 3: Heterogeneous Query Reasoning.** The system extends to settings where multiple database paradigms coexist (e.g., SQL, Cypher, GQL), requiring joint reasoning over source selection, query generation, and result integration.

This paper highlights multi-database query reasoning as a key extension of Text2Cypher, requiring new mechanisms for routing, decomposition, and integration beyond single-database settings.

## 2. Related Work and Motivation

Most existing text to query systems assume a single database as the execution environment [1, 2, 3, 4]. Recent work has started to relax this assumption and move towards more realistic settings.

**Beyond Single Database in Text2SQL**    For relational databases, recent Text2SQL work explores settings beyond a single database or single schema. One direction focuses on selecting relevant tables and schema elements in large databases. For example, Chung et al. [8] use external tools like BigQuery's SQL Code Assist [9] to identify relevant tables. Methods such as MURRE [10], T-RAG [11], and R2D2 [12] improve retrieval over large schemas through query rewriting, structured table organization, and fine-grained table decomposition. A second direction studies query decomposition within a single database, where complex questions are split into sub-questions that are mapped to different tables and columns. Methods such as DCTR [7] and DMRAL [13] explore building explicit table relationship graphs or using sub-question decomposition to generate the answer. A third direction considers open-domain and multi-database settings, where the target databases are not always explicitly provided. ABACUS-SQL [5] and LinkAlign [6] focus on selecting the correct database from a large pool and then identifying relevant schema elements through retrieval and query rewriting. Furthermore, recent work explores heterogeneous and cross-schema environments. While Daviran et al.[14] study query adaptation across databases with different schema structures, SQLake[15] addresses query execution over weakly structured data lakes. These works remain within the relational paradigm and do not address cross-language reasoning, such as Cypher vs SQL.

**Semantic layers**    Semantic layers are one approach for managing access across multiple data sources. A semantic layer is an intermediate layer between raw data sources and end-user applications. It defines shared business concepts, such as metrics, dimensions, and entities, on top of underlying tables or databases, so that users do not need to work directly with raw schemas or query languages [16, 17]. Building such a layer requires identifying data sources, designing the semantic model, and aligning it with business needs [16]. Source heterogeneity introduces challenges such as semantic correspondence across sources, entity matching, and global to local schema mapping [18, 19]. Recent work extends semantic layers using knowledge graphs and LLMs to map tables and columns into shared vocabularies across relational databases [20]. Other approaches build reusable semantic views or knowledge-graph-based abstractions to improve interpretability and querying [21, 22]. However, these approaches usually rely on upfront modeling of shared semantics and schema alignment, which can be costly to design and maintain.

**Overall gap**    Despite these advances, most Text2SQL methods still assume that each query is resolved within a single execution context. Semantic layers offer an alternative by enabling access across multiple sources, but they rely on explicit upfront modeling of shared concepts and relationships, rather than supporting query-time reasoning over independent systems.

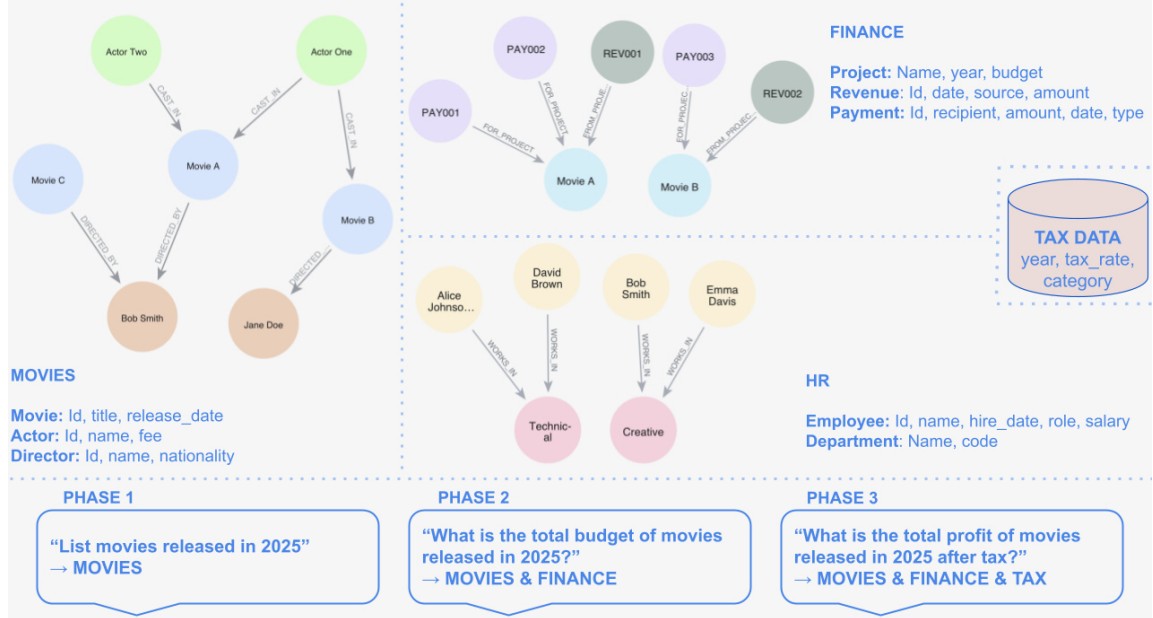

**Figure 1:** Example of multi-database query reasoning in a movie production company. Phase 1 uses a single graph database (Movies), Phase 2 combines two graph databases (Movies and Finance), and Phase 3 adds a relational database (Tax), combining Cypher and SQL results.

For the Text2Cypher task, to our knowledge, this problem remains largely unexplored. Existing systems assume a single preselected graph database and focus only on generating queries within that graph. They do not address scenarios where multiple graph databases must be considered, or they do not support routing, decomposition, or result integration across them. While some ideas from Text2SQL may transfer, graph databases introduce additional challenges. Schemas are defined implicitly through node labels and relationship types rather than explicit tables and foreign keys, and there is no native mechanism for querying across multiple graphs. As a result, reasoning across graph databases requires explicit coordination beyond standard query generation. This gap motivates our three-phase roadmap proposed in this paper.

## 3. Three-Phase Framework

In real-world settings, answering a single question may require accessing multiple independent graph databases, each with its own schema and structure. This requires the system to identify relevant sources, decide whether the question should be decomposed, and combine partial results without relying on a unified schema. We organize this problem into three phases:

- **Phase 1: Database Routing** handles routing to a single relevant database. Given input natural language question and pool of databases, the system first selects the relevant database and generates a Cypher query over it. The routing relies on semantic similarity between the question and database descriptions, schema representations or sample data.
- **Phase 2: Multi-Database Decomposition** extends routing to query decomposition and integration across multiple graph databases. Given an input question spanning multiple databases, the system first decomposes the question into sub-questions, assigns each to the appropriate database, generates per-database Cypher queries and aggregates the results.
- **Phase 3: Heterogeneous Query Reasoning** further extends the previous phases to heterogeneous settings where multiple query languages, such as SQL, GQL, Cypher, coexist. The system decomposes question, routes sub-questions to the appropriate databases and query languages and integrates the results across different data models.

**Illustrative Example** We illustrate the three phases using a movie production company with three independent databases (Figure 1): an HR database (employee information), a finance database (budgets and payments), and a movie database (production details). Each database has its own graph schema and no explicit cross-database links: **In Phase 1,** a question maps to a single database. For example, "List movies released in 2025" is routed to the movie database, where a Cypher query is generated. **In Phase 2,** a question may require multiple databases. For example, "What is the total profit of movies released in 2025?" requires retrieving movies from the movie database and financial data from the finance database. The system decomposes the question, generates queries for each database, and combines the results. **In Phase 3,** heterogeneous systems, where databases use different query languages, are involved. Extending the previous example with "after tax" introduces a relational database for tax data. The system must generate both Cypher and SQL queries and integrate results across different data models. Additional examples for the toy multi-database setting shown in Figure 1 are provided in Appendix A.

**Challenges and Open Problems** This setting introduces several challenges. **Benchmarks** for multi-database Text2Cypher does not exist. For the evaluation, new datasets with multiple graphs and cross-database questions are required. **Evaluation on routing** to the correct database is required, which may not be available in metrics currently used. **Evaluation on decomposition** of the input question also needs additional attention and new approaches checking validity even when multiple decompositions produce the same final answer is required. **Result integration** may require semantic alignment or additional techniques than just concatenation.

## 4. Conclusion

Most existing Text2Cypher systems assume a single preselected graph database and focus only on query generation within that context. However, real-world applications often involve multiple independent graph databases with no shared schema or direct connections. In this work, we propose a shift from single-database query generation to multi-database query reasoning, where routing, decomposition, and result integration are required in addition to query generation. We introduce a three-phase roadmap, composed of database routing, multi-database decomposition, and heterogeneous query reasoning, that extends current systems toward more realistic settings. Our analysis highlights that multi-database settings introduce new challenges that are not captured in single-graph benchmarks, including source selection, cross-database decomposition, and cross-system integration. This work formalizes multi-database reasoning for Text2Cypher and highlights key challenges for future systems.

## 5. Limitations

This work is a position paper proposing a research roadmap rather than a fully implemented system. The proposed phases are illustrated using a small set of hand-crafted examples and are not yet evaluated at scale on benchmark datasets or real-world multi-database workloads. The roadmap focuses on Text2Cypher, and does not empirically evaluate other query languages such as SQL, GQL. Finally, the multi-database setting assumes that all databases are known and accessible at query time, and does not address dynamic database discovery or evolving data sources. This assumption simplifies the routing problem and may not hold in open-world deployment scenarios.

## Declaration on Generative AI

During the preparation of this work, the author(s) used ChatGPT in order to: Grammar and spelling check, Paraphrase and reword, Improve writing style. After using these tool(s)/service(s), the author(s) reviewed and edited the content as needed and take(s) full responsibility for the publication's content.

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

**Table 1**
Illustrative examples of multi-database query reasoning across the three proposed phases.

| Phase | Question | Eval | Notes |
|---|---|---|---|
| **Phase1** | "List movies released in 2025" | ✓ | Identified the correct database (Movies) and produced a correct Cypher query. |
| | "Which department does Bob Smith work in?" | ✓ | Identified the correct database (HR) and produced a correct Cypher query. |
| | "How much is budgeted in total for 2025?" | ✓ | Identified the correct database (Finance) and produced a correct Cypher query. |
| **Phase2** | "What is the total budget of movies released in 2025? | ✗ | Identified only the Finance database, confused the `Project::date` field with `Movie::release_date`. |
| | "Which films has Emma Davis produced and what revenue did they generate?" | ✗ | Decomposed the question in a logical manner and correctly identified the Movies and Finance databases, but hallucinated a non-existing relationship `(:Person)-[:PRODUCED]->(:Movie)`. |
| | "Which employees were hired between the release dates of MovieA and MovieB?" | ✓ | Identified the correct databases (Movies, HR), produced correct Cypher queries, and proposed a valid integration strategy for the outputs. |
| **Phase3** | "What is the total profit of movies released in 2025 after tax?" | ✗ | Decomposed the question in a logical manner and correctly identified the Movies, Finance and Tax databases, but introduced ungrounded semantic mappings, such as associating movies with the tax category 'Entertainment'. |

data to knowledge graphs, arXiv preprint arXiv:2511.06455 (2025).

[21] A. Rissaki, I. Fountalis, N. Vasiloglou, W. Gatterbauer, Towards agentic schema refinement, arXiv preprint arXiv:2412.07786 (2024).

[22] P. Juhlin, R. Hussein, N. Schoch, Towards efficient field service engineering for powertrains via llm-generated knowledge graphs, SGKi 2025: Scaling Knowledge Graphs for Industry Workshop, co-located with SEMANTiCS 2025: International Conference on Semantic Systems (2025).

# Appendix

## A. Additional Examples on LLM Behavior

We provide additional illustrative examples for the toy multi-database setting shown in Figure 1. The goal is to analyze LLM behavior across the three phases, focusing on routing accuracy, decomposition quality, and cross-database reasoning errors. We use the Gemini-3.1-Flash-Lite model via Google AI Studio. The model is prompted with a schema-augmented instruction providing three Neo4j graph schemas (HR, Finance, Movies) and one relational schema (Tax), and is asked to generate queries and specify the target database for each query.

We evaluate representative questions from each phase. The results in Table 1 show that the model can generally perform database routing and produce meaningful decompositions. However, errors occur in cross-database settings, including hallucinated relationships, schema mismatches, and unsupported semantic assumptions during result integration. Overall, these examples indicate that while current LLMs handle routing and decomposition reasonably well in isolation, multi-database reasoning introduces distinct failure modes not observed in single-database Text2Cypher benchmarks.