# OpenReview forum: "Toward Multi-Database Query Reasoning for Text2Cypher"
_ijcai.org/IJCAI-ECAI/2026/Workshop/GENAIK-NORA — IJCAI-ECAI 2026 Joint Workshop on GENAIK and NORA_

### Official Review · Reviewer_rWL4 · 2026-05-17
**Good coverage of Text2SQL approaches (MURRE, T-RAG, R2D2, DCTR, DMRAL, ABACUS-SQL, LinkAlign), discussion of semantic layers as an alternative approach. Existing work assumes single execution context; this paper extends to multiple independent databases. Empirical validation consists of 6 hand-crafted examples with a single LLM showing 50% failure rate, with no error analysis or proposed solutions.**

**Rating:** 5
**Confidence:** 3

**Review:**

I appreciate the opportunity to review your manuscript on research roadmap for extending Text2Cypher systems from single-database to multi-database query reasoning titled, "Toward Multi-Database Query Reasoning for Text2Cypher". The work identifies a relevant gap in the literature existing Text2Cypher systems assume a single preselected graph database, while real-world deployments often involve multiple independent graph databases.

The three-phase roadmap is logical and well-structured

Good coverage of Text2SQL approaches (MURRE, T-RAG, R2D2, DCTR, DMRAL, ABACUS-SQL, LinkAlign), discussion of semantic layers as an alternative approach. Existing work assumes single execution context; this paper extends to multiple independent databases. Empirical validation consists of 6 hand-crafted examples with a single LLM showing 50% failure rate, with no error analysis or proposed solutions. The paper acknowledges these limitations but they are fundamental: it is not an empirical research contribution.

To Improve:
(1) Propose concrete algorithms for routing, decomposition, integration;
(2) Define phases formally with complexity analysis;
(3) Create benchmark dataset with multi-database queries;
(4) Implement baseline methods (naive routing, rule-based decomposition, etc.);
(5) Systematically evaluate multiple LLMs and prompting strategies;
(6) Analyze failure modes and propose solutions;
(7) Compare against adapted Text2SQL methods; (8) Provide ablation studies.

Accept for workshops/position tracks. Position paper identifying unexplored problem with three-phase roadmap. Suitable for workshops (Beyond-SQL, specialized venues)

---

### Official Review · Reviewer_Rp5i · 2026-06-02
**Review of paper 3**

**Rating:** 6
**Confidence:** 4

**Review:**

I read this position paper with interest. It identifies an under-explored gap in natural language interfaces: moving from single-database Text2Cypher generation to multi-database query reasoning. The author makes a fair point that while open-domain routing and multi-schema decomposition are gaining traction in Text2SQL research, graph databases present different hurdles. Because their schemas rely on node labels and relationship types instead of explicit tables and foreign keys, they require a completely different approach.

The paper maps out this problem using a clear three-phase roadmap. For an exploratory workshop submission, it does a good job of starting the conversation and setting a direction for future work. That said, it is strictly conceptual right now. I noticed it lacks a formal methodology and large-scale empirical validation.

Strengths:

- The focus on moving from single to distributed, multi-database environments captures a very real problem for enterprise applications.
- The three-phase roadmap accurately breaks down the escalating complexity of the task. This gives the community a practical starting point for building benchmarks and designing systems.
- The toy experiments using Gemini-3.1-Flash-Lite in the appendix are genuinely useful. They point out specific failure modes unique to this setting, like hallucinating cross-database relationships and creating ungrounded semantic mappings.

Weaknesses:

- The paper outlines the problem well, but it stops short of offering solutions. I was looking for a concrete architectural proposal to handle the deep retrieval, semantic alignment, or query decomposition needed to actually execute these phases.
- The current empirical evidence is limited to a few hand-crafted examples on a toy schema. To be fully convincing, the work needs a quantitative evaluation that measures routing accuracy, decomposition validity, or integration success.
- The framework assumes all candidate databases are known and accessible at query time. In actual multi-database setups, dynamic source discovery and changing schemas are major structural challenges. The author mentions these briefly, but they really should be integrated into the core phases rather than treated as an afterthought.

---

### Official Review · Reviewer_xanT · 2026-06-05
**Position paper for Question Answering over multiple / heterogeneous datasource**

**Rating:** 7
**Confidence:** 4

**Review:**

The paper presents a position/research roadmap for the problem of LLM-based question answering over potentially multiple independent databases. It argues that all existing works on text-to-SQL or text-to-Cypher/GQL present approaches and benchmarks that are based on a single database whereas the case where one wants to perform QA over multiple (potentially interconnected over semantic layers) datasources is not studied.

I would agree that the argument is correct and such an approach doesn't exist. One would need to create separate complementing datasources and somehow serve them via different endpoint and then at query time potentially perform query decomposition and federation to answer the question.

As a position paper/demo it is hard to assess novelty or contribution. Afterall it is at a proposal/idea stage. I think this idea is valuable from a business setting. The problem is related to data analytics over a data mesh where you might have different and potentially heterogeneous systems and want to perform analytics using cross-domain knowledge (finances + marketing + ... departments).

I find a bit confusing the use of wording of Phase 1/2/3. It more like these are increasing levels of complexity of the problem rather than phases in a step-by-step algorithm that solves the problems.

I would like to also know some more details on whether there is already some preliminary solution. E.g., has the author tried to do query decomposition? Did they use prompting?

---

### Official Review · Reviewer_gzDd · 2026-06-09
**Review of Submission 3**

**Rating:** 4
**Confidence:** 4

**Review:**

The paper argues that current Text2Cypher systems assume a single graph database; however, real-world deployments distribute data across multiple independent graph databases. It proposes reframing the task as multi-database query reasoning and organizes it into a three-stage roadmap: (1) database routing to a single relevant graph, (2) multi-database decomposition with per-database Cypher generation and result integration, and (3) heterogeneous query reasoning across coexisting query languages (SQL/GQL/Cypher). The setting is illustrated with a movie-production company comprising HR, finance, movie, and tax databases, and a small qualitative probe using one LLM on seven hand-crafted questions surfaces failure modes such as hallucinated relationships and schema mismatches.

Concerns:
1. The abstract and introduction repeatedly state that the work "formalizes" the setting and provides a "structured formulation," but the paper contains no formal problem definition, notation, objective, or evaluation criterion.
2. Phases 1 and 2 (routing and decomposition) are close to the multi-database Text2SQL works (e.g., ABACUS-SQL, LinkAlign, MURRE, DCTR). The novelty, the graph-specific difficulties, is identified but not developed into concrete mechanisms.

---

### Decision · Program_Chairs · 2026-06-10

Accept